# Stimulus–Secretion Coupling Mechanisms of Glucose-Induced Insulin Secretion: Biochemical Discrepancies Among the Canonical, ADP Privation, and GABA-Shunt Models

**DOI:** 10.3390/ijms26072947

**Published:** 2025-03-24

**Authors:** Jorge Tamarit-Rodriguez

**Affiliations:** Biochemistry Department, Medical School, Complutense University, 28040 Madrid, Spain; tamarit@ucm.es

**Keywords:** pancreatic islet metabolism, β-cell glycolysis, β-cell lactate release, β-cell OxPhos regulation, insulin secretion, β-cell secretion coupling mechanisms, canonical, ADP privation, GABA-shunt models, cell hierarchy of ATP consumer processes, β-cell mitochondrial ATP supply versus ATP demand, sulfonylureas and β-cell metabolism, ATP demand by insulin gene translation, β-cell mitochondrial alterations in T2D islets

## Abstract

Integration of old and recent experimental data consequences is needed to correct and help improve the hypothetical mechanism responsible for the stimulus–secretion coupling mechanism of glucose-induced insulin secretion. The main purpose of this review is to supply biochemical considerations about some of the metabolic pathways implicated in the process of insulin secretion. It is emphasized that glucose β-cells’ threshold to activate secretion (5 mM) might depend on the predominance of anaerobic glycolysis at this basal glucose concentration. This argues against the predominance of phosphoenolpyruvate (PEP) over mitochondrial pyruvate oxidation for the initiation of insulin secretion. Full quantitative and qualitative reproduction, except the threshold effect, of glucose-induced insulin release by a permeable methylated analog of succinic acid indicates that mitochondrial metabolism is enough for sustained insulin secretion. Mitochondrial PEP generation is skipped if the GABA-shunt pathway is exclusively coupled to the citric acid cycle, as proposed in the “GABA-shunt” model of stimulus–secretion coupling. Strong or maintained depolarization by KCl or sulfonylureas might induce the opening of β-cells Cx36 hemichannels, allowing the loss of adenine nucleotides and other metabolites, mimicking the effect of an excessive mitochondrial ATP demand. A few alterations of OxPhos (Oxidative Phosphorylation) regulation in human T2D islets have been described, but the responsible mechanism(s) is (are) not yet known. Finally, some experimental data arguing as proof of the relative irrelevance of the mitochondrial function in the insulin secretion coupling mechanism for the initiation and/or sustained stimulation of hormone release are discussed.

## 1. Introduction

The stimulation of insulin secretion by pancreatic β-cells is triggered physiologically by an increase of the plasma glucose concentration and by some other metabolites in in vitro conditions (initiators stimuli). Glucose-stimulated insulin secretion is biphasic. The first (triggering or initiating) phase is transient, and it reaches a peak value that lowers rapidly after 10 min to secretion rates above basal levels. Later, insulin secretion starts to rise progressively to reach steady levels (sustained phase), equal to or higher than the peak value of the first phase. Other metabolites and hormones behave as potentiator stimuli that need the presence of glucose to increase insulin secretion (fatty acids or gastrointestinal peptides). The glucose secretion mechanism is of a metabolic nature and is not mediated by glucose-sensitive receptors as it happens in other cellular types responsible for hormone and neurotransmitter secretion. The almost generally accepted canonical model of the metabolic glucose secretion coupling mechanism resides in the existence of at least five molecular processes in β-cells:A membrane glucose transporter of high capacity and low affinity (GLUT2, km~15–20 mmol/L) [1] constantly maintains the intracellular (cytosolic) sugar concentration in equilibrium with its plasma concentration.Glucokinase (km~4–10 mmol/L) phosphorylates glucose at a much slower rate than GLUT2-mediated uptake without affecting its distribution equilibrium across the plasma membrane [2]. The cooperation between GLUT2 and GK constitutes a glucose sensor that transduces variations of the glucose plasma concentration in proportional rates of its metabolism. Phosphorylated glucose is then metabolized in the known pathways of glycolysis and citric acid cycle, which leads to proportional rates of ATP synthesis in both pathways with predominance of ATP synthesized by the mitochondrial oxidative phosphorylation. This results in an increase in the cellular ATP/ADP ratio.Increases in the ATP/ADP ratio are sensed by ATP-dependent K^+^-channels (K^+^_ATP_) of the plasma membrane that respond in opposite directions to ATP and ADP concentrations. The prevalence of ATP closes the channels, whereas ADP opens them, as has been demonstrated. ATP closing of the K^+^_ATP_ channels suppresses K^+^-outward diffusion through the plasma membrane, decreases its electrochemical potential, and the membrane depolarizes, increasing its electrical potential to less negative values [3].Membrane depolarization activates a voltage-dependent Ca^2+^-channel that opens and allows the inward diffusion of Ca^2+^ ions, decreases its electrochemical potential, increasing the β-cell Ca^2+^ cytoplasmic concentration that occurs as oscillations of variable frequency and amplitude according to the degree of glucose stimulation [4].The mechanism of initiation of the insulin secretory response to glucose is subject to debate about the origin of the ATP supply required to close the K^+^_ATP_ channels by glycolysis or mitochondrial oxidative phosphorylation. This review aims to describe and compare how the different metabolic alternatives of either glycolysis and citric acid cycle support the three known hypothetical models (canonical, ADP privation, and γ-aminobutyrate (GABA)-shunt models) of the stimulus–secretion coupling mechanism of stimulation of insulin secretion. Two recent publications are recommended to anticipate a general knowledge of the ADP privation [5] and GABA-shunt [6] models.

## 2. β-Cell Glycolysis in the Canonical Model

### 2.1. Generality

The first enzyme in the glycolytic pathway is glucokinase (hexokinase IV), which due to its high Km, enables glucose phosphorylation in an extracellular sugar concentration-dependent manner [1]. Besides this highly specific feature, it is not indirectly regulated by variations of the cellular ATP as it occurs with other isoenzymes, like low Km hexokinases (HK I, II, and III), which are inhibited by its product, glucose-6-phosphate, which accumulates when glycolysis is suppressed by ATP inhibition of the enzyme phosphofructokinase, the main regulatory enzyme of glycolysis [2]. This is an optimal characteristic allowing a progressive increase of the β-cell glycolytic flux in response to an elevation of glucose independently of the cytoplasmic ATP levels, differentiating β-cells from most other body cells in which cytosolic ATP levels remain constant.

The fate of pyruvate, the product of glycolysis, has two alternatives: 1. Its reduction to lactate by lactate-dehydrogenase (LDH) that is released to the extracellular compartment (anaerobic glycolysis). 2. Aerobic glycolysis: pyruvate is transported to mitochondria by a specific transporter [7,8], where it is oxidatively decarboxylated to acetyl-S-CoA generating NADH by pyruvate dehydrogenase (PDH). Acetyl-S-CoA condensation with oxaloacetate (OxAc) by citrate synthetase then generates citrate, the first step in the citric acid cycle. OxAc supply is generated by an alternative carboxylation of pyruvate to OxAc by pyruvate carboxylase (PC), which requires an equimolar hydrolysis of ATP. This is an important “anaplerotic” reaction contributing to maintaining adequate levels of citric acid cycle intermediates to allow an efficient metabolic flux through the cycle. OxAc may also be supplied from alternative sources like amino acids or amines.

### 2.2. Islet LDH Activity

Lactate dehydrogenase (LDH) catalyzes pyruvate reduction to lactate coupled to NADH reoxidation in the cytoplasm. Its high K_eq_ constant favors lactate production. There are contradictions in the literature with respect to the relative importance of LDH activity in β-cells as compared with other competitor shuttles: the glycerophosphate dehydrogenase (GPDH) and the malate-aspartate shuttles. These are as follows:Homogenates of rat pancreatic islets have a high content of mGPDH activity, as compared with other tissues, and this is inhibited by diazoxide [9]. It was concluded that the GPDH shuttle had a key role in the stimulus–secretion coupling of glucose-induced insulin release.It has been reported that islet LDH activity is much higher than hexokinase-like activity measured at 1 mM glucose and lower than cytosolic glycerophosphate dehydrogenase [10]. It was hypothesized that “lactate production from glucose is not restricted by the activity of LDH but rather by the availability of its endogenous substrates, cytosolic pyruvate and NADH”. Due to the highest catalytic activity found in the cGPDH, it was suggested that the glycerol phosphate shuttle might allow an efficient reoxidation of the cytoplasmic NADH generated by GAPDH and maintain a high glycolytic flux.“LDH activity in isolated β-cells was shown to be 8-fold lower than in islet non-βcells and some 122-fold lower than in liver cells” [11]. The activity ratio LDH/mGPDH measured in primary β-cells was much smaller than in INS-1, HIT-T15, RINm5F, and liver cells. It was suggested that the low LDH/GPDHm ratio might ensure a tight coupling between mitochondrial metabolism and glycolysis in β-cells and that an increase in LDH expression or a decrease in GPDH might lead to a reduced secretory response to glucose.Overexpression of LDH-A in isolated MIN6 β-cells by intranuclear cDNA microinjection or adenoviral infection, “diminished the response to glucose of both phases of increases in mitochondrial NAD(P)H, as well as increases in mitochondrial membrane potential, cytosolic free ATP, and cytosolic free Ca^2+^” [12]. These effects were most pronounced at submaximal glucose levels. The insulin responses to an increase of the glucose concentration from 3 to 11 or 30 mM were considerably reduced. However, the response to a basal glucose concentration (3 mM) was unaffected. The authors concluded that mitochondrial pyruvate oxidation plays a critical role in the stimulation of insulin secretion in the β-cell. It might be also inferred that an increase in β-cell LDHA derives pyruvate metabolism from aerobic to anaerobic glycolysis.According to previous studies, β-cells poorly express LDH and the plasma membrane monocarboxylic acid transporter (MCT). Overexpression of LDHA and MCT-1 in INS-1 cells did not affect glucose-induced insulin secretion [13]. However, LDH overexpression allowed external 2 mM uniformly labeled lactate to be metabolized to ^14^CO_2_ and stimulate insulin secretion, independently of the co-overexpression of MCT-1. A total of 20 mM lactate stimulated insulin secretion in control and LDHA-overexpressing INS-1 cells, irrespective of the concomitant overexpression of MCT-1. The authors suggest that the low β-cell expression of LDH and MCT-1 might prevent excessive stimulation of insulin secretion by an increase of blood lactate during intense exercise; an unexpected pyruvate stimulation of insulin secretion might also be prevented in primary β-cells.LDH is expressed in mammalian cells as tetrameric proteins composed of all the possible combinations of two different M (muscle) and H (heart) subunits. In islet endocrine cells, the homomeric isozymes LDH1 (4H) and LDH5 (4M), codified by the LDHB and LDHA genes, respectively, predominate [14]. Isozyme LDHB is specifically expressed in β-cells and LDHA in α-cells [14,15]. LDHB can be inhibited dose-dependently with the specific inhibitor AXKO-0046, which lacks any effect on LDHA [14]. Isolated human and rodent islets transduced with the β-cell-specific lactate FRET sensor, Ad-RIP-Laconic, were stimulated by an increase of glucose concentration from 3 to 17 mM. The presence of 10 µM AXKO-0046 increased slightly (10–20%) but significantly the stimulation of lactate production by glucose. A total of 10 µM AXKO-0046 did not affect glucose stimulation of the ATP/ADP ratio in human β-cells but partially suppressed both the amplitude and frequency of glucose- and K^+^-stimulated Ca^2+^ oscillations. In contrast with AXKO-0046, galloflavin (10 µM), an inhibitor of both LDHA and LDHB, diminished glucose-stimulated lactate production in human β-cells [15]. None of the two LDH inhibitors used (AXKO-0046 and galloflavin) significantly modified 17 mM glucose stimulation of insulin secretion in human β-cells. In apparent discrepancy, 10 µM AXKO-0046 strongly stimulated human β-cell insulin secretion in response to basal (3 mM) glucose but galloflavin had no effect. Some LDHB expression quantitative trait locus (cis-eEQTLs) for decreased LDBH expression is associated with increased fasting insulin concentration [15]. The authors concluded that “LDHA is the major driver of glucose-stimulated lactate generation in human β-cells, and that LDHB limits this effect to maintain lactate within a tight range”.

### 2.3. Lactate Release by Isolated Rat Islets

Previous publications have reported high Km values for glucose induction of lactate release: 7 [16], 8.3 [17], and 15 mM [18]. The differences might be attributed to the use of different methods lacking the required specificity or sensitivity for measuring lactate. In collaboration with the Department of Histology and Medical Biology of Umeå University (Prof. J. Sehlin), we applied a modification of a previously published method [19] based on HPLC separation of a lactate derivative of high extinction coefficient, allowing lactate measurement in the picomolar range [20].

#### 2.3.1. Mesaurement of Lactate Release by Incubated Rat and ob/ob Islets

A wide range of glucose concentrations (0 to 20 mM) was assayed in parallel with lactate release, glucose utilization, and glucose oxidation by isolated and incubated rat and ob/ob islets [20]. There was a significant lactate release in the absence of glucose, which increased with the sugar concentration up to 3 mM and reached its half-maximal rate between 0.2 and 1.0 mM glucose in both species. The rates of lactate release and glucose utilization were similar at 3 mM glucose in rat islets and about 6 mM glucose in ob/ob islets. The observed rates of lactate release by rat and ob/ob islets at 3 and 20 mM glucose were almost identical, expressed in pmol/µ DNA x hour; the same similarity was observed with the corresponding rates of glucose utilization. These results support the idea that islet glucose metabolism at substimulatory concentrations is predominantly channeled to lactate. The results obtained in ob/ob islets support the interpretation of lactate release as the result of β-cell anaerobic glycolysis; ob/ob islets have a high proportion of β-cells [21].

#### 2.3.2. D-Manoheptulose (MH)

A known specific competitive inhibitor of glucokinase inhibited dose-dependently lactate release and glucose utilization in both rat and ob/ob islets [20]. A close-to-maximum effect was reached at 5 mM MH. The suppression of islet lactate release by MH suggests that at least part of lactate production is generated by glucokinase; however, the participation of a low Km hexokinase cannot be discarded.

## 3. β-Cell Glycolysis in the ADP Privation Model

The ADP privation model proposes that glycolysis generates enough cytosolic ATP to close the K^+^_ATP_ channel by the pyruvate kinase (PK) reaction, without the participation of oxidative phosphorylation. This enzyme transfers the energy of the energy-rich phosphoenol bond of PEP to ADP to synthesize ATP and release pyruvate. The experimental core supporting this model was the finding that PEP addition to β-cell plasma membrane excised patches in the inside-out configuration in the presence of ADP-blocked K^+^-currents. As a corollary, it is suggested that there must be a metabolon with glycolytic activity at the plasma membrane close to the K^+^_ATP_ channels [22].

Cellular PK activity is exerted by three enzymatic isoforms: PKM1, PKM2, and PKL [22,23]. Their enzymatic activity was measured in recombinant copies of the three isoforms [22]. While PKM1 is constitutively active, the other two isoforms are allosterically activated by fructose-1,6-bisphosphate (FBP) and several pharmacological drugs; both types of activators lower the Km of PKM2 for PEP [23]. According to Lewandowski et al. [22], TEPP-46 (10 µM, PKa), one of the known PKM2 activators, lowered the Km of global PK activity in lysates of INS-1 cells from 1.6 to 0.11. It increased PKM2 enzymatic activity in a range of PEP concentrations (0 to 2.5 mM). PKM1 activity was several-fold more active than PKM2 and was not affected by PKa [22].

INS-1 cells responded dose-dependently to glucose (2.5 to 16.7 mM), and the presence of 10 µM PKa significantly potentiated insulin secretion at all the glucose concentrations, including the substimulatory ones, in INS1 832/13 cells and, apparently, also in human donor islets [22]. To our opinion, PKa behaves similarly to a GK activator (0.5 nM RO-0281675, GKa): it induced a smaller decrease of the EC50 but also induced a much higher maximum secretory response than GKa. The authors concluded that “although PK is sufficient to close K_ATP_ channels, the lack of a threshold effect makes it unlikely that PK activation enhances insulin secretion by increasing fuel intake”. To our opinion, PKa mimics the allosteric activation of PKM2 as it is a “regulatory enzyme” (Newsholme E.A., Start C., JohnWiley&Sons, Chapter 1, page 15) [24] that might accelerate the glycolytic flux at the last step despite a lowering of its substrate PEP.

## 4. Different Glycolysis Function in the Postulated Models of Stimulus–Secretion Coupling Mechanism of Glucose-Induced Insulin Secretion

Substimulatory concentrations of glucose lower than 5 mM are mainly metabolized by anaerobic glycolysis as supported by the quantitative coincidence of the rate of glucose utilization with the rate of lactate production. These experimental data suggest that the omission of mitochondrial pyruvate metabolism does not allow sufficient synthesis of ATP to close K^+^_ATP_ channels and imposes a threshold for the initiation of insulin secretion by glucose. At substimulatory glucose concentrations, glycolysis is also generating PEP, perhaps increasing its cytosolic concentration. Its failure to stimulate insulin secretion might be attributed to the requirement of a higher PEP concentration at a higher glycolytic rate (above 5 mM glucose) and/or to the cooperative synthesis of mitochondrial PEP in the citric acid cycle once exceeded the rate of pyruvate reduction to lactate. Above 5 mM glucose, both aerobic glycolysis and the citric acid cycle might be coupled and capable of generating both cytosolic and mitochondrial PEP. This idea seems compatible with the canonical and GABA-shunt models of the stimulus–secretion coupling mechanism of glucose-induced insulin release but poses a caveat for the idea that glycolysis is fully responsible for the closure of K^+^_ATP_ channels by glucose in the ADP privation model.

The cytosolic reduction to lactate competes with pyruvate transport through the inner mitochondrial membrane, mediated by a mitochondrial pyruvate carrier (MPC). Therefore, LDH activity might limit mitochondrial pyruvate transport.

Mammal MPC is a heterodimer of two monomers, MPC1 and MPC2 [25]. They catalyze the transfer of cytosolic pyruvate together with 1H^+^, and, therefore, its transport is favored by a pH gradient between the cytosol (7.2) and mitochondrial matrix (8.0), 0.8 being a physiologically relevant value. Published values of the Km recorded in the literature for MPC and LDHB (A) are in the millimolar (1.1 ± 0.4 mM) and micromolar range, (64 µM) respectively [8,14]. These biochemical data support the contention that pyruvate reduction predominates at lower pyruvate (and glucose) concentrations versus mitochondrial pyruvate transport.

Islet β-cells have a poor expression of monocarboxylate transporters (MCT) located in the plasma membrane [26]. The MCTs family consists of 14 members, and the first four isoforms, MCT1–4, catalyze the proton-coupled transport of monocarboxylates and play an important role in cell metabolism [26]. The immunodetection of MCT1, the most common in many tissues, was almost zero in β- and α-cells, and it was absent from MIN6 cells and present in low amounts in INS-1 and at high levels in RINm5F cells. Consequently, β-cells are not, or poorly, stimulated by exogenous lactate [13]. However, the co-overexpression of LDHA and MCT1 allowed the stimulation of insulin secretion by pyruvate and lactate in rat islets. Whatever the MCT isoform predominating in β-cells is, even though expressed at low levels, it may be mediating the efflux of lactate. One must be cautious about the fact that some of the known inhibitors of MCTs also impact mitochondrial pyruvate transport (MPT), like α-cyano-4-hydroxycinnamate (CHC) [27], 7ACC2, and lonidamide [25].

2.Another argument supporting the above caveat (paragraph 1 above) is that the “allosteric” activation of PKM2 and PKL by a specific activator (TEPP-46 or PKa) lowers their Km and increases their enzyme activity [22]. In our opinion, PKa activation should be able to increase the glycolytic flux at the last step, despite lowering PKM2 substrate (PEP), as corresponds to a “regulatory enzyme” [24]. The same mechanism explains the beneficial effect of GK activators on insulin secretion in T2D β-cells [28]: as commented in the previous text, the GK activator (piragliatin) increased several metabolic islet parameters at 3 mM glucose. As already known, caution is needed to prevent intense hypoglycemia if the referred enzyme activators are used to treat T2D.

## 5. Islet Mitochondrial Metabolism in the Canonical Model

### 5.1. Secretory Activity of Exogenous Pyruvate on Insulin Secretion

Pyruvate failed to stimulate insulin secretion by rat islets up to a 30 mM concentration in the absence of glucose, but it exhibited some secretagogue activity at higher concentrations (60 and 90 mM) [29]. In the presence of 8.3 mM glucose, 30 mM pyruvate shifted the dose–response curve of glucose stimulation of insulin secretion to the left.

Another old report concluded that glycolysis is fully responsible for the closing of β-cell K^+^_ATP_ channels whereas mitochondrial ATP is irrelevant for the stimulation of insulin secretion by glucose in mouse islets due to the sub-compartmentation of ATP within the β-cells [30]. The main argument provided was that glucose (12 mM) stimulation of insulin secretion was not suppressed by α-cyano-4-hydroxycinnamate (CHC), nor by fluoroacetate. CHC was used as a supposed inhibitor of mitochondrial pyruvate transport (MPT) but it is a more specific inhibitor of plasma membrane monocarboxylate transporters (MCT) at the low concentrations used (1 mM) [26]. Therefore, CHC exerted no suppression on methyl pyruvate (plasma membrane permeable analog)-stimulated insulin secretion. Surprisingly, fluoroacetate (inhibitor of the citric enzyme aconitase) neither inhibited the secretory response to 12 mM glucose, whereas respiratory chain inhibitors (rotenone and antimycin A) induced a strong suppression. An experimental confirmation of fluoroacetate inhibition of glucose oxidation would have been appreciated.

Pyruvate methyl ester (20 mM) stimulated dose-dependently (5 to 40 mM) a biphasic insulin secretion in perifused rat islets [31]. The first phase of secretion was of similar magnitude to that triggered by 20 mM glucose but the second, the sustained phase, was six-fold smaller. Sodium pyruvate, in the absence of glucose, did not stimulate insulin secretion.

### 5.2. Secretory Activity of Succinic Acid Methyl Esters on Insulin Secretion

The succinic acid membrane permeable analogs mono (SAM) and dimethyl (SAD) esters were found to be insulin secretagogues in rat pancreatic islets, but succinic acid itself was inactive [32,33]. SAM stimulated dose-dependently (1 to 20 mM) insulin secretion and, at 10 mM, both SAM and SAD triggered an equivalent insulin response in incubated islets [32]. ^14^C-labeled dimethyl succinate is metabolized to CO_2_ by pancreatic islets but not ^14^C-labeled succinic acid [34], confirming that succinic acid does not permeate the plasma membrane. Dimethyl[1,4-^14^C] succinate (labeled SAD) oxidation to CO_2_ was potentiated by 20 but not 1 mM glucose; a similar effect was induced by 1 mM acetate but not 1 mM pyruvate. It is tempting to speculate that the lack of effect of pyruvate might be due to its low membrane permeability because of the low expression of MCT transporters, as commented above. Protonated acetate is probably more permeable than pyruvate and may give rise to an increased mitochondrial acetyl-CoA concentration that would mimic the effect of pyruvate dehydrogenase in the absence of glucose.

It has been demonstrated that succinic acid methyl esters behave as insulin secretagogues in human and rat but not mouse islets [35]. The author attributes the failure of mouse islets to their deficiency of “malic enzyme” expression. This cytosolic enzyme, also known as “malate dehydrogenase (oxaloacetate decarboxylating)”, catalyzes the following reaction: malate + NADP^+^ → pyruvate + CO_2_ + NADPH. It participates in the shuttle to transport acetyl-CoA from the mitochondria to the cytosol for lipid synthesis and provides pyruvate that may be decarboxylated in the mitochondrial matrix to acetyl-CoA by pyruvate dehydrogenase to favor the complete oxidation of citric acid cycle intermediates like succinic acid. This shuttle may explain why exogenous acetate supplies pyruvate to favor dimethyl[1,4-^14^C] succinate oxidation, as mentioned above [34].

SAM (20 mM) alone stimulates a biphasic insulin response characterized by an initial peak followed by a small but significant sustained phase [36]. At a substimulatory glucose concentration (2.75 mM), the sustained, but not the initial, phase was amplified around four-fold. D-Mannoheptulose (20 mM) did not interfere with the SAM secretory response.

The dynamic insulin repose to SAM, in the absence of glucose, was compared to that exerted by glucose at equal concentrations (10 and 20 mM) [37]. The first phase of the biphasic secretion stimulated by 10 mM SAM was very similar to that triggered by 10 mM glucose. However, the magnitude of the sustained phase of insulin secretion due to 10 mM SAM was only 25% the size of the one triggered by 10 mM glucose. A similar difference was confirmed at 20 mM concentrations of both SAM and glucose.

Another report supplied evidence that SAM and SAD stimulated a poor biphasic secretion of insulin in the absence of glucose that was strongly potentiated by the presence of 6.0 mM glucose [38]. Both methylated succinic acid analogs showed a very similar dose–response curve for stimulation of insulin secretion.

A report from our laboratory compared the stimulation of insulin secretion by SAD and glucose in perifused rat islets [39] to investigate the relative contributions of glycolysis versus oxidative metabolism to the stimulus–secretion coupling mechanism of β-cells. SAD stimulated biphasic insulin secretion in a dose-dependent manner in the absence of glucose. A total of 3 mM SAD already triggered a clear biphasic insulin response that was bigger at 7 and 10 mM. Higher SAD concentrations, 13 and 20 mM, did not trigger a greater response. A total of 10 mM pyruvate alone did not elicit any secretory response; however, it potentiated the insulin responses to 10 mM SAD and 20 mM glucose by 58 and 64%, respectively. The secretory response to 20 mM glucose alone was almost identical, both qualitatively and quantitatively, to that triggered by 10 mM SAD alone. The perifusion of 10 mM SAD together with 20 mM glucose induced a 60–70% enhancement of insulin secretion with respect to the separate responses triggered by 10 mM SAD and 20 mM glucose separately.

Dimethyl malonic acid (MAD) is a permeable analog of malonic acid that inhibits competitively the citric acid cycle enzyme succinic acid dehydrogenase. It suppressed dose-dependently the secretory response of perifused islets to 10 mM SAD and 10 and 20 mM glucose. A total of 10 mM MAD decreased the rate of islet oxidation of 10 mM [2-^14^C]pyruvate and 20 mM [U-^14^C]glucose by 30 and 71%, respectively, but had no effect on glucose utilization [39].

## 6. Islet Mitochondrial Metabolism in the ADP Privation Model

The ADP privation model postulates that PEP synthesis in the citric acid cycle may also have a contribution in the cytosol in closing the K^+^_ATP_ channels in a similar way to glycolytic PEP during the sustained phase of glucose-induced insulin secretion.

The enzyme phosphoenolpyruvate carboxykinase (PEPK) is expressed in mammalian cells as two isoforms, cytosolic (PEPCK-C) and mitochondrial (PEPCK-M), that catalyze the decarboxylation of oxaloacetate (OxAc) into phosphoenolpyruvate (PEP) as a cataplerotic derivation of the citric acid cycle or as a flux generating reaction of gluconeogenesis in the cytosol, according to the following reactions, respectively:

OxAc + GTP → PEP + CO_2_ + GDP + Pi and OxAc + ATP → PEP + CO_2_ + ADP + Pi

PEPCK-M mRNA is 15-fold higher than PEPCK-C mRNA in INS-1 832/13 cells and 11-fold higher in rat islets [40]; therefore, β-cell gluconeogenesis is practically inoperative and PECK-M is a source of PEP whose metabolic role was unknown in β-cells until recently. The GTP required for mitochondrial PEP synthesis is supplied by the previous citric acid cycle enzyme, succinyl-CoA synthetase, which hydrolyzes the thiol ester bond of CoA in the presence of GDP + P_i_ and releases succinic acid + GTP [41]. The stoichiometry of GTP and PEP is 1 per citric acid cycle turn. The resultant GTP does not leave mitochondria but PEP is in equilibrium with the cytosolic PEP pool, which also includes glycolytic PEP generated by enolase from 2-phosphoglyceric acid [40,41,42].

PEPCK-M silencing in INS-1 cells with two different siRNAs strongly suppressed both its mRNA and protein expression [40]. PEP labeling with 5 mM [3-^13^C]pyruvate at 15 mM glucose, as well as insulin secretion, were also strongly suppressed. PCK2^−/−^ mice showed worse glucose tolerance in OGTTs, and knockout mouse islets had a reduced first and second phase of insulin secretion stimulated by 16.7 mM glucose than control wild-type islets [43]. This important finding partially supports the “ADP privation model” of glucose–secretion coupling of insulin release.

Intracellular PEPCK-M flux was determined by measuring the label incorporation into PEP from 5 mM [3-^13^C]pyruvate in INS-1 832/13 cells that have a high expression of plasma membrane monocarboxylate transporter. In the presence of a range of glucose concentrations (0, 3, 7, and 15 mM), the percentage of total (labeled + unlabeled) mitochondrial PEP relative to glycolysis increased dose-dependently. At a substimulatory (3 mM) glucose concentration, the percentage of mitochondrial PEP supply was significantly increased compared to the absence of glucose [40]. [U-^13^C]gucose labeling of INS1 832/13 cells’ metabolites followed by mass spectrometry showed that labeled PEP content was increased by a range of glucose concentrations (2.5, 5.0, 9.0, and 16.7 mM). Statistical comparisons among glucose concentrations were missing [22].

An increase of 10 to 13 mM glucose increased significantly the duty cycle and period of both the ATP/ADP ratio and cytosolic [Ca^2+^] in human islets that were significantly reduced by a lowering of glucose concentration (from 10 to 8 mM) [22]. The additional presence of GKa did not induce, apparently, any extra effect to the stimulation by 10 mM glucose (no statistical confirmation); by contrast, PKa decreased significantly both the calcium and ATP/ADP ratio duty cycles and periods at 10 mM glucose. In summary, PKa did not potentiate, but unexpectedly suppressed glucose effects on calcium and ATP/ADP ratio oscillations, despite its capacity to potentiate the stimulation of insulin release. The addition of PKa significantly suppressed cellular PEP labeled content at 2.5 and 5.0 mM [U-^13^C]glucose and only increased it at 16.7 mM labeled glucose. The oxygen consumption rate (OCR) measured in mouse islets and INS-1 cells at 2.5 and 9 mM glucose was not affected by 10 µM PKa but it was suppressed by oligomycin and rotenone.

PKa was shown to increase mitochondrial NADH fluorescence and induce a strong hyperpolarization of the mitochondrial membrane potential (ΔΨ_M_) at 2 and 9 mM glucose in mouse islets [22]. According to the authors, these experimental results “would be consistent with a decreased OxPhos”. However, in our opinion, these experimental data are indicative of a mitochondrial state III respiration that disagrees with the lack of OCR stimulation by PKa on islets and INS-1 cells, and that is expected to increase oxidative phosphorylation. PKa-induced increase of both NADH and ΔΨ_M_ at a substimulatory glucose concentration, like 2.5 mM, strengthens the idea that it also stimulates basal insulin secretion.

The authors [22] tested whether PK itself is capable of limiting OxPhos (Oxidative Phosphorylation) by ADP privation in situ in permeabilized (XF Plasma Membrane Permeabilizer) INS-1 832/13 cells in the presence of 10 mM succinate: exogenous ADP (62.5, 125, and 250 µM) dose-dependently stimulated state III respiration, and the extracellular addition of a range (68 to 625 µM) of PEP concentrations caused an instantaneous and similar, non-concentration dependent, suppression of mitochondrial respiration. It is difficult to understand why the experiments were performed with succinic acid instead of glucose. First, succinic acid alone does not stimulate secretion in control non-permeabilized cells, according to the authors’ experimental data. Second, the target of PKa is the glycolytic PK but there is no evidence that succinic acid alone might increase cellular PEP. A total of 10 mM Phe (an inhibitor of PK) [40] was shown in an older publication to increase significantly the islet content of PEP, although it did not significantly modify glucose-induced insulin secretion [44].

An earlier study from our laboratory studied the secretory capacity of exogenous PEP (10 mM) on perifused and permeabilized (70 mM KCl) rat islets; it induced a biphasic insulin secretion with a modest second phase as compared with 20 mM glucose (in control non- permeabilized islets) [45]. The insulin response was not affected by 10 µM rotenone but it was strongly suppressed by 15 mM Phe (a PK inhibitor). However, rotenone suppressed the ATP content in incubated non-permeabilized islets by almost seven-fold. Our permeability procedure by 70 mM KCl opened only β-cell Cx36 hemichannels that allow the loss (or exchange with the extracellular space) of molecules below 1 KD, including many metabolites like adenine nucleotides. Direct addition of exogenous metabolites like PEP in our permeabilized islets escapes cellular metabolic control; it is not synthesized by the mitochondria, and therefore, it is not physiological. It just served to know which glycolytic or mitochondrial metabolite was more efficient in increasing intracellular ATP and directly stimulating insulin secretion. Therefore, the failure of rotenone to suppress exogenous PEP-induced secretion in permeabilized islets lacks any physiological interpretation. This same argument may be applied to the experiments described in the above paragraph by Lewandowski S. L. et al. [22]. An experimental confirmation that PKa, in the presence of glucose instead of succinate, increases the intracellular PEP concentration simultaneously to the increase of both, mitochondrial ΔΨM and NADH is missing. However, Phe suppression of extracellular PEP-induced insulin secretion in permeabilized islets supports the possibility that PK-dependent PEP hydrolysis might contribute to the stimulation of insulin secretion in non-permeabilized islets.

## 7. Functional Metabolic Alternatives of the Citric Acid Cycle and OxPhos in the Three Models of the Stimulus–Secretion Mechanism of Glucose-Induced Insulin Secretion

Succinic acid dimethyl ester (SAD) reproduces qualitatively and quantitively the effect of a maximum stimulus of glucose despite skipping glycolysis. However, SAD, at variance with glucose, did not show any threshold effect for stimulating insulin secretion. It is also evident that succinic acid is not a physiological stimulus of insulin secretion, but it supports the idea that direct stimulation of mitochondrial metabolism may trigger a full stimulation of insulin secretion.Our previous proposal of the required use of the GABA-shunt for the oxidative metabolism of glycolytic-derived pyruvate [6] is compatible with the canonical model of the stimulus–secretion coupling of glucose-induced insulin release. However, the GABA-shunt pathway skips the synthesis of PEP by the GTP-dependent succinyl-CoA synthetase. As the “ADP privation” model [22] requires the mitochondrial supply of PEP to sustain the stimulation of insulin secretion by glucose, it would be interesting to know how much the limitation exerted by the low expression of α-ketoglutarate dehydrogenase (αKGdh in Figure 1) allows a significant citric acid cycle flux in the two pathways. It has been demonstrated in INS-1 832/13 cell line and rat islets [46,47] that the intracellular concentration of α-ketoglutarate (αKG or 2-OG) is increased more than 20-fold (more intensively than any other mitochondrial metabolite investigated) at 16.7 mM, as compared with 2.8 mM glucose, strongly supporting a strict control of the cyclic acid cycle flux by the αKGdh in β-cells.Even though it is out of the scope of this review, three possible mechanisms responsible for the oscillating behavior of sustained insulin secretion have been proposed. It is debated whether they are attributed to oscillations in the ATP/ADP ratio driven by ADP regulation of mitochondrial OxPhos (ADP privation model) [5], to cytosolic Ca^2+^ changes inducing cyclic variations of β-ell bioenergetics (canonical model) [48], or to K^+^_ATP_ conductance oscillations “driving bursting electrical activity and pulsatile insulin secretion” [49].

## 8. Comments on Some Assertions About the Relative Irrelevance of OxPhos in the Initiation of Glucose Stimulation of Insulin Secretion

The assertion that “β-cells do not metabolize glucose below the circulating concentration set point of 4–5 mM under fasting conditions” [50] is not supported by experimental data showing that glucose utilization is almost fully accounted for by lactate production in the same range of glucose concentrations [20].

CHC was used as a supposed inhibitor of mitochondrial pyruvate transport (MPT) to demonstrate that inhibition of the mitochondrial pyruvate metabolism had no effect on glucose or methyl pyruvate insulin release [30]. However, there is experimental evidence that CHC is a more specific inhibitor of plasma membrane monocarboxylate transporters (MCT1 and 2) at the low concentrations used (1 mM) [27].

Pyruvate dehydrogenase kinase (PDK) activity phosphorylates the E1α subunit of PDH blocking its enzymatic activity; its inhibition might reactivate PDH and accelerate pyruvate oxidation in the citric acid cycle. Abulizi et al. [43] argued that knockout of PDKs 2 and 4 should activate PDH activity but it was not corroborated experimentally; the authors support their assertion on a reference by Wu CY et al. [51] who studied the effects of PDK2 and PDK4 knockout on glucose homeostasis and liver metabolism, but not in islets, of diet-induced obese mice. Abulizi A. et al. [43] showed that dichloroacetate (DCA, a PDK inhibitor) unexpectedly and significantly suppressed 9 mM and 16.7 glucose stimulation of insulin secretion in both rat islets and INS-1 cells, as well as in islets from mice with a double-knockout of PDHKs 2 and 4. Phosphoenolpyruvate carboxykinase-knockout mice (Pck2^−/−^) exhibited a significantly greater glucose AUC than control mice following an OGTT that was not corrected by the infusion of PKa. The significantly defective insulin response of Pck2^−/−^ mice subject to an OGGT by PKa was neither reversed. All these experimental data do not support the idea that increased glucose (pyruvate) oxidation has no effect on the β-cell secretory response.

Akhmedov D. et al. [52] demonstrated that DCA blocked the phosphorylation of PDH E1α Ser^293^ in INS-1E cells. Specific siRNA knockout of PDHKs 1, 2, and 3 markedly suppressed glucose-induced E1α Ser^293^ phosphorylation without modifying basal or glucose-stimulated release, nor the amplitude of the glucose-induced mitochondrial Ca^2+^ rise. A total of 16.7 mM glucose reduced PDH activity to 70% of the total, but it decreased [1-^14^C]pyruvate oxidation to ^14^CO_2_ by 30% compared to 3 mM glucose. The authors conclude that “the observed PDH E1α phosphorylation does not impact on metabolism secretion coupling”. According to these experimental data, the DCA inhibitory effect on glucose stimulation of insulin release is not attributable to inhibition of β-cell PDK activity. It seems more plausible that DCA acts also as an inhibitor of mitochondrial pyruvate transport (MPC), as has been reported previously [53,54].

## 9. OxPhos Function Regulation and Its Participation in the Stimulation of Insulin Secretion

Tamoxifen-inducible β-cell specific knockouts of the respiratory complexes CI, CIII, and CIV “had similar mitochondrial respiratory defects [55]. CIII knockout defects were the most diabetogenic; they caused early hyperglycemia, glucose intolerance, and loss of glucose-stimulated insulin secretion in vivo”. “Paradoxically”, 16.7 mM glucose stimulation of insulin secretion by perifused islets of CIII KO mice was apparently normal. There was also a small difference (*p* < 0.02) in the percentage of glucose cells responding with mitochondrial [Ca^2+^] increases to a 16.7 mM glucose stimulus between control and CIII KO cells. However, the biphasic dynamics of insulin secretion exhibited an almost complete loss of the second phase even in the controls. The percentage decrease (with respect to the basal rate) of islet respiration in response to 17 mM glucose was not as straightforward as expected from a complete knockout of the respiratory complexes; moreover, there were no significant differences in the relative (in percentage of the basal rate) rate of maximal respiration among the control and the three groups of knockout islets. The authors suggest that an increased anaerobic glycolysis rate might compensate for the suppressed rate of mitochondrial respiration by an elevated expression of the following mRNAs codifying for enzymes of glycolysis or other related enzymes: Glut2, GK, LDH-A and B, and the plasma membrane monocarboxylate transporter (MCT). However, according to some antecedents commented above, an increased rate of anaerobic glycolysis by LDHA overexpression [12] suppresses the stimulation of insulin secretion in the low glucose concentration range. The finding of downregulated expression of OxPhos complexes in islets from T2D patients suggests that mitochondrial respiration also plays some relevant role in the stimulus–secretion coupling mechanism of glucose stimulation of insulin secretion.The use of “sea horse” has uncovered that islet mitochondrial respiration is less coupled to OxPhos than in myocytes [54]. Rat INS1 cells and human islets have a lower degree of respiration uncoupling (around 38%) than mouse islets (between 49 and 59%). Myocytes have a smaller uncoupling degree (23%) than human islets and INS1 cells. The authors consider it counterintuitive that the ATP content of INS1 cells is much higher (more than five-fold) than in myocytes. They conclude that this striking difference might be attributed to the different physiological roles of β-cells (“fuel-sensors”) and myocytes (“fuel utilizers”). It is unknown which is the cellular mechanism regulating mitochondrial uncoupling in β-cells; there is recent evidence that the uncoupling protein 2 (UCP2) does not participate [56]. Perhaps the better the bioenergetic efficiency the higher the cellular ATP content, or the lower the metabolic cost for cellular survival (anaplerosis) the higher the ATP reservoir (cataplerosis). In other words, in our opinion, β-cell regulates its ATP production by ATP supply rather than by ATP demand [5].Gerencser A. A. et al. [57] developed a model of fluorescent probe dynamics allowing the quantitation of the mitochondrial membrane potential (ΔΨ_M)_) in a monolayer culture of individual cells. Δp represents the difference of an electrochemical potential whose value depends on two components: a chemical gradient of [H^+^] (ΔpH) and an electrical potential (ΔΨ_M_) between matrix and cytosol (Figure 2). Its magnitude depends on the balance between the respiratory efflux of H^+^ to the cytosol by the respiratory complexes and their return to the matrix mediated by the F1F0-ATP synthase that transduces the energy accumulated in ΔΨ_M_ into the synthesis of ATP from matrix ADP + Pi (oxidative phosphorylation, OxPhos) or due to proton leakage (any uncoupling mechanism yet unknown). The regulation of cellular bioenergetics depends on the interaction between three mitochondrial modules: glucose oxidation (measured as oxygen consumption rate (ORC), phosphorylation (ATP synthesis), and proton leak (uncoupled respiration) [50,57].

Gerencser A.A. et al. [59] applied the abovementioned new protocol in cultured INS-1 832/13 β-cells to study the interaction between the three mentioned modules controlling the bioenergetics of β-cells. An increment of glucose concentration from 2 to 10 mM caused a gradual increase in ΔΨ_M_ from −118 ± 3.9 to −141 ± 4.1 mV in 10 min. In isolated mitochondria, the maximal rate of ATP production approximately doubles with each 10 mV increase in ΔΨ_M_. A total of 2.5 µM oligomycin (FoF1-ATP synthase inhibitor) caused a drop to a lower rate of uncoupled respiration. The latter was dose-dependently inhibited by rotenone (respiratory complex I inhibitor) or increased by FFCP (trifluoromethoxy carbonyl cyanide phenylhydrazone, a proton transporter (ionophore) that shortcuts the H+- flux down the FoF1-ATP synthase suppressing the synthesis of ATP). Both FFCP and rotenone decreased ΔΨ_M_. The OCR recordings at 2 and 10 mM glucose in the absence of oligomycin were higher than in its presence, indicating the existence of a positive feedback loop between the “phosphorylation module” (ATP/ADP or other downstream signals such as Ca^2+^) and OCR (enzymes controlling glucose oxidation like glucokinase or phosphofructokinase). As OCR measures coupled and uncoupled (proton leak) respiration in the absence of oligomycin, the authors concluded that glucose stimulates both the rate and the “coupling efficiency” of oxidative phosphorylation. This means that “most of the glucose stimulated respiratory rise is coupled to oxidative phosphorylation while the proportion of basal respiration used to make ATP is exceptionally low” [50]. One wonders whether the priority of anaerobic over aerobic glycolysis might be responsible for the lower proportion of coupled respiration at basal glucose concentrations (<5 mM). As a corollary, “control of ΔΨ_M_ is exclusively exerted by glucose oxidation, a process that is regulated by (yet unknown) factors that arise downstream from ATP synthesis” [56]. This “supports the notion that glucose-sensing by the β-cells is not accounted for by glucokinase alone”. This concept is further strengthened by the fact that 5 mM pyruvate, or its methyl derivative, in INS-1 and primary cells, respectively, stimulate respiratory activity to the same extent as 20 mM glucose, suggesting that “nutrient-sensing is at least partially controlled by mechanisms downstream from glycolysis” [50]. Gerencser A.A. et al. [56] conclude that “increased metabolism of glucose and not inhibition of futile cycling in glycolysis or altered ATP demand mediates hyperpolarization of ΔΨM upon an increase of glucose concentration”.

Cellular energization, ΔΨ_M_ hyperpolarization, and ATP/ADP are subject to strict homeostatic regulation in many tissues like the liver and heart due to a balance between cellular ATP supply and ATP demand that maintains the ATP/ADP content constant [56]. By contrast, the pancreatic β-cells are supposed to exhibit, according to the canonical model of the stimulus–secretion coupling mechanism of glucose-induced insulin secretion, a lack of homeostatic preservation of the ATP/ADP that varies with the glucose concentration. Homeostasis of the glucose concentration is achieved by an adequate release of insulin in response to a metabolic demand (hyperglycemia) of the organism. It seems reasonable that β-cells behave as “fuel sensor” cells whose predominant function is to increase their insulin secretion by incrementing their ATP content. Therefore, β-cells bioenergetic regulation must be more dependent on “ATP supply”. This is compatible with minor participation of β-cell ATP demand in the regulation of its cellular bioenergetic system [50].

## 10. Control of ATP Flux by ATP Demand

The energetic consumption and dependence of the main ATP-demanding cellular processes on OxPhos [60] have been measured in some mammalian cells (categorized as “fuel utilizers”). The protocol followed was to measure the decrease in OCR induced by the specific inhibition of the different ATP-demanding processes individually. The results uncover a bioenergetic hierarchy in which the higher O_2_ (ATP) consumers were protein and RNA/DNA synthesis, followed by sodium cycling through the Na^+^/K+-ATPase, Ca^+^ ATPase, unidentified ATP consumers, and proton leak. “Together these identified processes account for 94% of the respiration of the cells” [57]. Titrating OCR progressively with myxothiazol (CRIII inhibitor), a scale of dependence of each ATP-demanding (-consumer) process on OxPhos could be determined by examining the decay of each individual process with the inhibition of OCR. The following assumption is made: “inhibition of one ATP consuming process does not stimulate the rate of the others through increases in cellular ATP levels”. All these consumer processes are in turn driven by the energy accumulated as mitochondrial membrane potential (ΔΨ_M_) after its transduction to ATP (OxPhos). OxPhos controls 88% of the total flux through ATP, 20% through substrate oxidation, and 8% through proton leak. Correlatively with the previously defined hierarchy, protein and polynucleotide synthesis were the most sensitive to the ATP supply (30 and 21%, respectively). Sodium and calcium ATPase consumed 14 and 15%, respectively. In our modest opinion, there is not yet any exhaustive investigation of the β-cell bioenergetic needs. What is the position of the variable consumption of the exocytotic mechanism? Is there a need for a “leak” or any other mechanism to adapt on time the secretory ATP supply to the ATP demand? Is the high ATP content of β-cells (fuel “sensor” cells), compared with other “utilizer” cells, a reservoir allowing a fast triggering of insulin secretion once the cytosolic Ca^2+^ increases?Inhibition of mRNA translation
(a)A total of 80 µg/mL puromycin (a translation inhibitor) suppressed by 95% ^14^C-valine incorporation into total islet protein that was unaffected by the reduction of extracellular calcium; insulin synthesis was assumed to be decreased in a similar percentage [61]. After 30 min exposure of perifused islet to 80 µg/mL puromycin, they were stimulated with 16.7 mM glucose in the absence and presence of 80 µg/mL puromycin. The latter depressed substantially the second phase of insulin secretion. The calculated percentage contribution of newly synthesized to the total insulin released was around 30%. The secretory response of perifused islets to 16.7 mM glucose was directly proportional to the perifusate Ca^2+^ concentration up to 4 mM. Tolbutamide (around 40 mg/mL) triggered a first phase of secretion and an almost absence of a second phase in the absence or presence of a substimulatory glucose concentration. It would be interesting to know whether variations of extracellular Ca^2+^ influence islet OCR.(b)Inhibition of mRNA translation by 5 µM cycloheximide suppresses acutely β-cell respiration in INS-1E cells and mouse islets stimulated by 11 and 28 mM glucose, and the suppression is reversed by an uncoupler of mitochondrial respiration [50]. As expected, cycloheximide suppression of protein synthesis decreases the corresponding ATP demand, and the ATP supply should also be proportionally reduced to reach a new steady state. It is unknown whether this balance is caused by a reduced glucose metabolism or is due to negative feedback by an alteration of the ATP/DP ratio. The fact that neither cycloheximide nor puromycin might exert their effects through unexpected translation inhibition of other proteins cannot be discarded.

## 11. Control of ATP Flux by Sulfonylureas

Sulfonylureas are known to potentiate glucose-induced insulin secretion through the closing of K_ATP_ channels and are currently used for the treatment of patients with type 2 diabetes.

(a)Tolbutamide (0.74 mM) inhibited ^86^Rb^+^net uptake and increased ^22^Na^+^uptake in rat islets in the absence of glucose. It also significantly increased the uptake of ^45^Ca^2+^ [62]. The authors attributed the effect to β-cell depolarization. Might the tolbutamide effect be due to an off-target action of the sulfonylurea?(b)A range of tolbutamide concentration (3–500 µM) was shown to potentiate insulin secretion by a range of glucose concentrations (5.10, 15, and 30 mM) in mouse islets. The sulfonylurea predominantly increased the first phase of insulin secretion. The second phase declined slowly to less than 60% of the peak rate after 10 min of islet perifusion. Tolbutamide’s half-maximum effective concentration was 10 to 30 µM [63]. Islet OCR was significantly increased by 100 µM tolbutamide in a range of glucose concentrations (from 5 to 30 mM). Tolbutamide (30 or 100 µM) lowered islet ATP content significantly at 5 or 10 mM glucose. This might be an example of increased ATP demand at low glucose (5 mM) that cannot be compensated by an increased ATP supply. However, it does not explain the tolbutamide suppressive effect at higher concentrations. Might the cause be related to an off-target effect of plasma membrane depolarization?(c)A rise of the glucose concentration from 1 to 10 mM glucose increased concomitantly the secretion of insulin and the ATP/ADP ratio in previously cultured mouse islets [64]. A total of 2 µM nimodipine (an inhibitor of voltage-dependent Ca^2+^ channels) increased the ATP/ADP ratio in a range of stimulatory glucose concentrations (over 6 mM). This unexpected effect of blocking voltage-dependent Ca^2+^ channels is difficult to reconcile with the expected stimulation of mitochondrial metabolism by a rise of the cytoplasmic Ca^2+^ concentration after glucose stimulation.

In contrast, islet depolarization with either 30 mM KCl or 100 µM tolbutamide unexpectedly decreased the ATP/ADP ratio at sub- (3 and 5 mM) and stimulatory (10 and 30 mM) glucose concentrations. The suppressive effect of the ATP/ADP ratio by 30 mM KCl at 10 mM glucose was not modified in the presence of 100 µM diazoxide (blocker of the K^+^_ATP_ channels). The authors conclude that “Ca^2+^ exerts a feedback control of K^+^-ATP channels via changes in ATP consumption and hence of the ATP/ADP ratio”. However, this argument does not explain why the increase of cytosolic Ca^2+^ concentration by glucose exerts the opposite effect on the ATP/ADP ratio as KCl or tolbutamide even though they increase insulin release. Has depolarization with KCl or nimodipine any off-target effect on the bioenergetics of the β-cell?

(d)10 µM glibenclamide significantly suppressed islet ATP and ADP contents in ob/ob islets at 0 mM glucose, and 10 mM glucose abolished its suppressive effect [65]. It had no effect on glucose oxidation in a wide concentration range of sulfonylureas (0.1, 1.0, and 10.0 µM) and glucose concentrations (1.0, 10.0, and 20.0 mM). A total of 20 mM glucose increased the ouabain-sensitive but suppressed the ouabain-resistant ^86^Rb^+^ uptake. Glibenclamide increased the ouabain-sensitive 5 min ^86^Rb^+^ uptake at 1.0 and 10.0 µM concentrations in the absence of glucose, but there was no additive effect with the addition of 20 mM glucose [66]. Glucose did not stimulate Na^+^-K^+^-dependent ATPase in islet homogenates [67]. A total of 1 mM ouabain suppressed progressively in time the ATP content of ob/ob islets at different glucose concentrations (0, 3, and 20 mM); after 30 min of ouabain exposure, the islet ATP content was significantly decreased at all the assayed glucose concentrations [68]. Similar results were obtained by 30 mM KCl at 3 and 20 mM glucose. The authors suggest that the sulfonylurea-induced decrease of islet ATP content is due to the (indirect?) stimulation of the Na^+^-K^+^-dependent ATPase by means of the resultant plasma membrane depolarization. If this cannot be confirmed by measuring the glibenclamide effects on mitochondrial ΔΨM, perhaps one might explore a possible off-target effect of the sulfonylurea. In current terms, sulfonylureas seem to decrease OxPhos by an increase in the ATP demand [68].(e)In β-HC9 cells, the sulfonylurea glyburide, at 5 mM glucose, increased OCR to a minor degree than 30 mM glucose but induced a greater insulin response [69]. ^31^P-NMR provides estimations of phosphocreatine (PCr) and inorganic phosphate (Pi) in live cells (ADP levels are below the detection limit), and PCr/Pi is a measure of energy state that is correlated with the ATP/ADP/Pi. A study of the ^31^P-NMR spectra of superfused β-HC9 cells showed that 1 µM glyburide, at 5 mM glucose, decreased cellular phosphocreatine and increased inorganic phosphate levels without affecting ATP content; opposite changes were induced by 30 mM glucose. Similar results were obtained in a ^23^Na-NMR spectroscopy study: 5 and 30 mM glucose decreased cellular cytosolic Na^+^, whereas 1µM glyburide, at 5 mM glucose, increased it substantially. Is sulfonylurea-induced Na^+^ loss caused by a decrease in cytosolic ATP?(f)In rat islets, 1 µM of the sulfonylurea glibenclamide, at 10 mM glucose, had little effect on cytochrome c reduction, increased OCR, and decreased the “calculated” ATP/ADP/Pi levels by 50% [70]. The authors interpret that glibenclamide ATP suppression is not due to a restriction of substrate availability (poor reduction of cytochrome c by the sulfonylurea as proof) but to a secondary increase of the energy demand triggered by an increased insulin secretion. It is difficult to understand that such a strong suppression of islet adenine nucleotides might be due to a huge increase of any ATP demand processes at a relatively high glucose concentration. One cannot discard that the repeated demonstration of a lowering effect of ATP and the ATP/ADP ratio by sulfonylureas or KCl depolarization might not be exclusively attributed to their capacity to increase insulin secretion.

Added comments

β-cell OxPhos may be necessary for the correct functioning of glucose activation of the stimulus–secretion mechanism of insulin secretion. As commented in the previous text, sulfonylureas (tolbutamide, glibenclamide, and glyburide) suppressed islet ATP content at low, and apparently less effectively, higher glucose concentrations. A similar effect was induced by depolarization with 30 mM KCl at 3 and 20 mM glucose. Our laboratory demonstrated that KCl or the omission of extracellular Ca^2+^ induces a loss of ATP from β-cells from rat and mouse islets at substimulatory glucose concentrations due to the opening of Cx36 hemichannels [71,72] resulting in a subsequent inhibition of glucose-induced insulin secretion at higher concentrations. The opening of Cx36 hemichannels was prevented by stimulatory glucose concentrations with an EC50% of 8 mM [71]. One wonders whether sulfonylureas and KCl-mediated depolarization of the β-cell plasma membrane might induce a derangement of β-cells after a long-term treatment. Might they mimic the effects of an excessive ATP demand for the regulation of mitochondrial OxPhos?

Besides primary β-cells, Cx36 (gene Gjd2) is also expressed in β-cell lines of human (EndoC-βHs) and murine (Min 6 and INS-1derivative 832-13) origin, but their expression degree is much lower [73,74]. Cx36 is very determinant for the electrical coupling of clonal β-cells within their aggregates (3D spheroids), enhancing their stimulus secretion coupling mechanism by improving the signal-to-noise ratio. Three-dimensional spheroids may provide surrogates of pancreatic islets for their functional study as a functional syncytium [74].

## 12. Reported Alterations of the Bioenergetic Control in T2D β-Cells and Islets

(a)Gerencser A.A. [75] performed a bioenergetic study on single β-cells from healthy and T2D humans. An increase of glucose concentration from 3 to 16 mM glucose induced consistently smaller effects on ΔΨP (variation of plasma membrane potential) and ΔΨ_M_ (variation of inner mitochondrial membrane potential) in T2D than in control β-cells: ΔΨ_P_ was less depolarized and ΔΨM was less hyperpolarized in T2D than in control β-cells. However, secretagogue mixtures of metabolites directly metabolized in the citric acid cycle (methyl-succinate + α-ketoisocaproate, and glutamine + BCH) hyperpolarized ΔΨ_M_ to the same extent in T2D as in control islets. A study of the relationship, or dependence, between ΔΨ_P_ and ΔΨ_M_ using glucose as the unique stimulus demonstrated no difference between the ΔΨ_P_ values of T2D and control cells corresponding to identical values of ΔΨ_M_ in both types of cells. This means that plasma membrane depolarization (ΔΨ_P_ less negative) is not affected in T2D cells. As confirmation, glibenclamide increased a similar ΔΨ_P_ depolarization (higher than that induced by glucose), corresponding to the same value of ΔΨ_M_, in both types of cells.

At 3 mM glucose and amino acids, 1 µM oligomycin hyperpolarized ΔΨ_M_ in both control and T2D β-cells, but at 16 mM glucose, only T2D β-cells mitochondria were hyperpolarized [75]. The authors suggest that “in normal β-cells substrate oxidation driven by high substrate availability determines ΔΨ_M_ with little sensitivity to variations in ATP demand when glucose is high. In contrast, diabetic β-cells are more sensitive to ATP demand, responding to oligomycin by mitochondrial hyperpolarization under identical conditions”. However, it is difficult to understand how an excess of the ATP demand over ATP supply might increase ΔΨ_M_ depolarization that is evoked by increased availability and rate of substrate oxidation (glucose or amino acids). Would it be possible that T2D cells have a larger contribution of an unknown proton leak in the group of mitochondrial ATP consumer processes than control cells?

A total of 50 µM glibenclamide, at 3 mM glucose and amino acids, depolarized ΔΨ_P_ more intensively in T2D than in control β-cells without modifying ΔΨ_M_ [75]. ΔΨ_P_ depolarization should induce hyperpolarization in ΔΨ_M_ due to an increased (glucose or other nutrients) metabolism stimulated by an increase of the cytosolic Ca^2+^ concentration. The lack of a ΔΨ_M_ hyperpolarization in T2D β-cells leads to the conclusion, contrary to the one reached with 16 mM glucose, that the diabetic cells are less sensitive to ATP demand than control cells. Caution should be taken regarding the fact that the stimulus used, “absence of elevated nutrient levels”, does mean a low concentration of nutrients: at 3 mM glucose alone, the rate of glycolysis is too low or only anaerobic, and then energetically poor, impeding an increase of mitochondrial metabolic flux that could hyperpolarize the membrane mitochondrial potential (Ψ_M_).

Anello M. et al. [76] comparatively studied the secretory and metabolic activities of isolated islets from healthy and T2D diabetic humans. Incubated diabetic islets showed a significantly smaller secretory response to 16.7 mM glucose than control healthy islets. T2D islets also exhibited a significantly higher ATP content than control islets at 3.3 mM glucose that failed to be increased by glucose, resulting in a lower ATP/ADP ratio. Mitochondrial ΔΨ_M_ hyperpolarization by a change of glucose concentration from 3.3 to 16.7 mM was 32.3% lower in T2D islets than in controls; the uncoupling rate after the addition of 1 µM FCCP was apparently smaller in diabetics than in control islets. In fact, T2D islets exhibited a higher expression of UCP2 and the respiratory complexes CI and CV. The authors concluded that the elevated UCP2 expression in T2D islets is responsible for the lack of response of the ATP/ADP ratio by stimulating glucose.

(b)Human T2D islets have a lower expression of the OxPhos gene set than healthy control islets [77]. Rosengren A. H. et al. [78] “identified four risk alleles that associate with impaired β-cell exocytosis and enabled us to form a novel genetic risk score for single β-cell dysfunction that involves impaired granule docking and defective Ca^2+^ sensitivity of exocytosis”. These two reports show that the cause of the development of T2D diabetes may be multifactorial and not only restricted to OxPhos impairment.(c)A report by Doliba N.M. et al. [28] demonstrates that piragliatin (RO4389620) is an allosteric GK activator that lowers the Km of the enzyme and increases OCR, usage, oxidation, and cytosolic Ca^2+^ concentration by/of glucose in a range of concentrations shifted to the left in mouse, rat, and human islets. A total of 3 µM piragliatin rescued the suppressed OCR, the stimulation of insulin secretion, and the cytosolic Ca^2+^ concentration in response to a staircase stimulus of glucose in T2D islets. What is impressive is that the GK activator was able to potentiate significantly OCR, insulin secretion, and cytosolic Ca^2+^ at substimulatory glucose concentrations. This experimental finding suggests that a defect in ATP supply, by diminished glucose metabolism, may cause a “diabetic conversion” of islet function. Caution should be taken with this GK activator for the treatment of T2D patients due to its potential to cause hypoglycemia.

All the described alterations of the bioenergetic regulation in T2D β-cells emphasize the importance of β-cell mitochondria in the homeostasis of β-cell function. Future experimental work is required to uncover the responsible mechanisms that might orient their prevention or treatment.

## Figures and Tables

**Figure 1 ijms-26-02947-f001:**
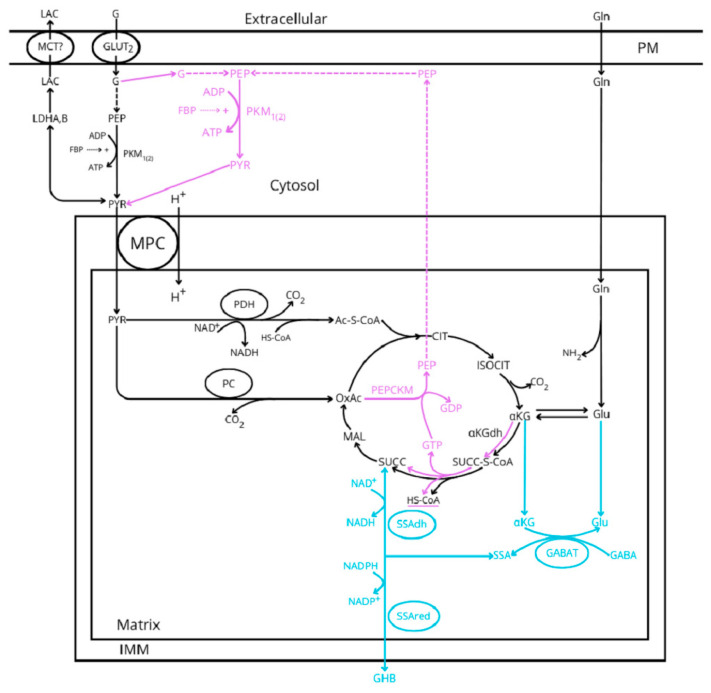
Glucose metabolic pathways implicated in the stimulus–secretion coupling of insulin release according to the three hypothetic models proposed: canonical and common (black lines), GABA-shunt (blue lines), and “ADP privation” (purple lines). MCT represents the monocarboxylic acid transporter, MPC is the mitochondrial pyruvate transporter, GABA is the gamma-aminobutyric acid, GHB is the gamma-hydroxybutyric acid, G is glucose, Lac is lactate, GLUT2 is the glucose transporter 2, FBP is the fructokinase bisphosphate, LDH is the lactate dehydrogenase, PK is the pyruvate kinase, PDH is the pyruvate dehydrogenase, PC is the pyruvate carboxylase, PEPCK is the PEP-carboxykinase (suffix M and C is mitochondrial or cytosolic), GABAT is the GABA transaminase, SSAdh and SSAred are the semialdehyde succinic acid dehydrogenase and reductase, respectively, PM is the plasma membrane, and IMM is the inner mitochondrial membrane. For the sake of simplicity, the outer mitochondrial membrane (OMM) has not been represented.

**Figure 2 ijms-26-02947-f002:**
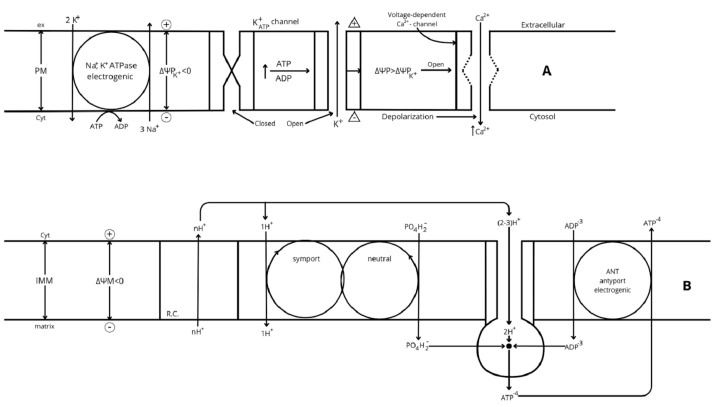
(**A**) Regulation of plasma membrane electric potential (ΔΨP = ΨP_Cyt_ − ΨP_ex_). The K^+^ diffusion gradient through the plasma membrane is at equilibrium with the membrane electrical potential in resting cells: the Na+, K+-ATPase catalyzes a non-compensated antiport of 2K^+^ in and 3 Na^+^ out of the cell, generating an inverse gradient of both ions. The higher plasma membrane permeability to K^+^ than Na^+^ allows a higher outwards diffusion of K^+^ than the inwards return of Na^+^, developing an increasing electric potential (ΔΨP) through the plasma membrane. In resting cells, ΔΨM hyperpolarizes until the influx of K^+^ promoted by the electrical potential counteracts its efflux rate by the generated concentration gradient, and the net K^+^ flux through the plasma membrane becomes null; the value of this electrical potential is called K^+^ equilibrium potential (ΔΨP_K_^+^ < 0). β-cell activation by ATP/ADP closure of K^+^_ATP_ channels diminishes the efflux of K^+^ without affecting the entrance of Na^+^ and induces a depolarization of the resting (or ΔΨP_K_^+^) potential, which becomes less negative. The Na+, K+-ATPase restores the altered ionic gradients after the stimulus. (**B**) Overview of the mechanism of oxidative phosphorylation. The specific potential redox created by each respiratory complex (RCs) is first transduced to an electrochemical potential by a proportional efflux of H^+^ from the matrix to the intermembrane space. The electrochemical potential generated (Δp = −ΔΨ_M_ + 60 (pH_m_ − pH_i_), also known as proton motive force (PMF), has two components according to its name: a concentration potential generated by the difference of pH between the matrix (m) and the intermembrane space (i) (ΔpH = pH_m_ − pH_i_ > 0), and an electrical potential (ΔΨM = ΨM_m_ − ΨMi < 0). The electrical component is the more directly implicated component, via hyperpolarization, in an increase of mitochondrial metabolism and the corresponding increase of oxygen consumption (OCR): its hyperpolarization is implicated in the final transduction of electrical potential into chemical energy for the synthesis of the terminal phosphodiester bond of inorganic phosphate (PO_4_H^2−^) to ADP to produce ATP (oxidative phosphorylation, OxPhos). The electrogenic antiporter (ANP) of matrix ATP^−4^ outward against cytosolic ADP^−3^ inward, favored by ΔΨM hyperpolarization, maintains a cytosolic ATP/ADP ratio greater than the mitochondrial one (around seven-fold in the liver) [58]. This means that the energetic value of ATP hydrolysis in the cytosol is greater than it would be in the mitochondria. On the other hand, the mitochondrial synthesis of ATP requires less energy than the reverse reaction. Symport of PO_4_H^−2^ with 1H^+^ is also energized by hyperpolarized ΔΨ_M_. Finally, the NADH/NAD^+^ ratio is greater in the mitochondrial matrix than in the cytosol, suggesting that cellular mitochondrial metabolism is steadily well supplied with substrates (m means mitochondrial matrix and i means intermembrane space).

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
