# Peer review of "Stimulus–Secretion Coupling Mechanisms of Glucose-Induced Insulin Secretion: Biochemical Discrepancies Among the Canonical, ADP Privation, and GABA-Shunt Models"

_ijms, 2025, doi:10.3390/ijms26072947_

Round 1

Reviewer 1 Report

Comments and Suggestions for Authors

This is a detailed review of biochemical mechanisms of glucose-stimulated insulin secretion (GSIS) in pancreatic beta cells. The paper focused on three mechanisms, and introduced a large amount of supporting and conflicting data. Since GSIS is quite complex and sophisticated, this review is very helpful for our understanding of this mysterious phenomenon.

1)Most of the readers are not so well-versed in GSIS or beta cell biology. For example, the biphasic insulin secretion, which is one the specific features of GSIS, should be described briefly. I also wonder which phases of GSIS are more associated with each model.

2)There are a lot of differences in GSIS among mouse, rat and human, and also between primary beta cells and cell lines. For example, the second phase of GSIS is less obvious in mouse islets compared with rat or human islets, and most of the beta cell lines showed impaired GSIS especially after long passages. This is quite important, since data from different sources of beta cells were introduced. Please just comment on this point in the earlier part of the review.

3)Since the various pathways or shuttles can work even at sub-stimulatory concentrations of glucose, I am keen to know whether these models may also contribute to the ‘permissive effects’ of glucose which contribute to the action of incretins. I understand this is not the main scope of this review, but just briefly mention this point in order to draw the attention of readers.

4)Minor point: Figure 1 GABA shunt is not highlighted in green but in blue.

Author Response

Thank you very much for your review, please find the responses in the attachment

Reviewer 2 Report

Comments and Suggestions for Authors

Review ijms-3424897: ‘Stimulus-Secretion Coupling Mechanisms of Glucose-Induced Insulin Secretion: Biochemical Discrepancies Among the Canonical, ADP-Privation, and GABA-Shunt Models’

This paper appears to be a GPT-generated report, primarily quoting various sources without sufficient critical analysis. Furthermore, it is largely based on the review published in Metabolites (2023, 13(6), 697; https://doi.org/10.3390/metabo13060697). Notably, the legend in Figure 1 states’ ‘The rest of biochemical acronyms are indicated in the text or in reference (39)’ demonstrating that the figure itself is largely adapted from figures in that reference.

This review lacks novelty and scientific rigor. Additionally, it does not fulfill the criteria of a scientific review, which requires critical analysis, synthesis, and integration of findings from multiple primary research studies to provide a comprehensive understanding of the subject. Some suggestions to improve this paper are outlined below, but similar changes should be applied throughout the manuscript.

________________________________________

Major Comments:

1. A schematic representation comparing the roles of hexokinases I, II, and III with hexokinase IV (glucokinase) would improve clarity. The expression levels of these kinases in β-cells compared to other tissues are not discussed, which is an important aspect of glucose metabolism.

2. The phrase: “The fate of pyruvate, the product of glycolysis, has two alternatives” is misleading. The discussion should frame the metabolic fate of pyruvate in the context of aerobic vs. anaerobic metabolism, rather than presenting these pathways as binary alternatives.

3. Regulation of Gluconeogenesis and Lipogenesis (Line 98):

The statement: “In some tissues, PC is the first step to gluconeogenesis; it also contributes to initiating the pathway of lipogenesis (synthesis of fatty acids and complex lipids from glucose).” lacks appropriate metabolic ( catbolic vs. anabolic)  context. The review should address that gluconeogenesis (in the liver) and lipogenesis (in the liver, adipose tissue, and other sites) are dynamically regulated by the body's energy status (fed vs. fasted states) and dietary composition (e.g., low-carbohydrate diets).

4. Aerobic vs. Anaerobic Conditions in Lactate Production (Chapter 3):

The section "Lactate production from glucose" should explicitly differentiate between aerobic and anaerobic conditions, as this distinction is fundamental to understanding metabolic flux.

5. NADH Shuttle Systems (Line 109):

The phrase “transfer of NADH from cytoplasm to mitochondria” is inaccurate and misleading. NADH shuttle systems do not physically transfer NADH or NAD⁺ but facilitate electron transfer to maintain redox balance between cytosolic and mitochondrial NADH/NAD⁺ pools. The glycerophosphate dehydrogenase (GPDH) and malate-aspartate shuttles should be described in terms of their role in balancing these redox states rather than in terms of NADH transport. 

Clarification of Shuttle Types (Lines 121 & 129): The statement “the net flux of the shuttle favors the reoxidation of cytoplasmic NADH by mitochondrial FAD” is unclear, since three different shuttle systems are discussed, it is necessary to specify which shuttle is being referenced in each instance. Similarly, the hypothesis regarding LDH activity being substrate-limited by cytosolic pyruvate and NADH availability (Line 129) should be contextualized in relation to aerobic vs. anaerobic metabolism.

6. LDH/mGPDH Ratio and Consideration of LDH Isoenzymes (Line 135): The discussion on the LDH/mGPDH activity ratio did not consider the five LDH isoenzymes (LDH1–LDH5), which have distinct tissue distributions and kinetic properties. Later the author mention LDHA and LDHB without any explanation. Furthermore, oxygen tension in β-cells, which differs from that in other tissues, should be discussed as a factor influencing LDH activity.

________________________________________

Minor Comments:

1. English Editing & Clarity Improvements:

The phrase: “Analysis of precedent and new data should allow now and then to suggest modifications for a future integral view” should be revised for clarity, for example:

“Data analysis may occasionally drive paradigm changes.”Similar ambiguous phrasing should be corrected throughout the manuscript.

2. Terminology Adjustments:

o “The main purpose of this review is to supply only biochemical considerations” – This statement should be more precise.

o “Full quantitative and qualitative reproduction” – Likely intended to mean reproducibility.

o “Some experimental data argued as proof of the relative irrelevance” – The meaning is unclear and requires rewording for precision.

3. Missing Citations in Introduction (Lines 31-50):

o This section lacks necessary references to support key claims.

4. Glucose-Secretion Mechanism (Lines 35-36):

o The statement: “The glucose-secretion mechanism is of a metabolic nature and is not mediated by glucose-sensitive receptors like it happens in other cellular types responsible for hormone or neurotransmitter secretion” contradicts the well-established role of GLUT2 in glucose sensing.

o This sentence needs rewording to accurately describe the role of glucose transporters and metabolic coupling factors in insulin secretion.

5. Unreferenced Statements:

o “As demonstrated time ago” (Line 51) and “debate” (Line 58) are not referenced and require citations.

6. Introduction Structure (Line 68): The final sentence of the Introduction should clearly state the focus and objectives of the review.

7. Exocytotic Machinery (Line 71):

o The phrase “exocytotic machinery that results in the extracellular release of insulin” is unclear.

o This likely refers to insulin secretion and should be revised accordingly.

o Additionally, this claim lacks proper referencing.

8. Terminology in Glycolysis and TCA Cycle Discussion (Line 74):

o The phrase “Glucose is majorly metabolized in the traditional pathways of glycolysis and citric acid cycle” should replace “traditional” with “catabolic” to better reflect metabolic terminology.

9. Glucokinase Description (Line 75):

o The phrase “glucokinase (hexokinase IV) that, thanks to its high Km” should avoid colloquial language like "thanks to" and use scientific phrasing, such as:

“Glucokinase (hexokinase IV), due to its high Km, enables glucose phosphorylation in a concentration-dependent manner.”

10. Formatting & Typographical Errors:

Subtitle issue: “2.Β-. cell glucose metabolism” – Incorrect formatting of β-cell.

Missing citations: Statements throughout Chapter 2 lack references, making it unclear which claims are supported by experimental data.

Typos:

o “2H, FAD” (Line 117) – Should be 2H⁺, FAD⁺.

o “FDAH2” (Line 118) – Should be FADH₂.

Inconsistent formatting in Chapter 3.

Use of direct quotations in a scientific review is atypical. The manuscript should summarize multiple experimental findings rather than directly quoting other authors.

Comments on the Quality of English Language

 Poor use of English see' minor comments'.  However, the major concern is about the format of this report, which does not integrate scintific work but provide a list of published work.

Author Response

Thank you for your review, please find the responses in the attachment

Reviewer 3 Report

Comments and Suggestions for Authors

This study provides an in-depth review of the biochemical features of the three models of stimulus-secretion coupling in glucose-induced insulin secretion. It is a valuable review that compiles various intriguing findings. However, the content is overly extensive and somewhat unorganized, making revisions necessary for better clarity and structure.

- Full names should be provided for all terms upon first mention, and abbreviations thereafter. In abstract several words did not have full name but only abbreviations (e. g. GABA), although several words had their full- name (e. g. PEP). 

- The introduction contains a substantial amount of specific data (e. g. Km = 10~40 mM glucose); however, there is a lack of supporting references. Please add appropriate citations to substantiate the statements (e.g. line: 40-48).

- Although this is a review paper, the detailed descriptions of results from references are overly complex (line: 123-186, 733-753, etc.) and several results were repeated, making the manuscript difficult to read. To improve readability, key points should be concisely summarized before providing specific examples and numerical data to support them. 

For example, (line: 353-363, line 487-489, even with figure number of references?) are overly detailed. It would be better to briefly introduce what MAD is and explain that it inhibits islet secretory function by targeting succinic acid dehydrogenase in the TCA cycle, along with a concise summary of the findings. Currently, the section includes excessive details from the original experimental study, including specific numerical data, which significantly reduces readability and disrupts the logical flow of the review.

- The sequence and flow of the review paper are difficult to follow. While the introduction lists key points in a numbered format (1–5), the order does not seem to align with the presentation of results. To improve clarity, consider structuring the introduction to provide a clearer guideline for the results section. Alternatively, including a graphical abstract at the beginning to outline the sequence of topics covered in the main text could enhance readability. 

- Would it be possible to categorize the overall content based on the canonical, ADP-privation, and GABA-shunt models mentioned in the title?

- It is hard to understand 'partial conclusion'. part. It should be more concise and title seems to be changed. 

Author Response

Thank you very much for your review, please find the responses in the attachment.

Reviewer 4 Report

Comments and Suggestions for Authors

This manuscript aims to provide a comparative analysis of the canonical, ADP-privation, and GABA-shunt models of glucose-induced insulin secretion. The topic is highly relevant to the field of pancreatic β-cell metabolism, and the authors attempt to challenge established models by presenting alternative perspectives. While the manuscript provides an extensive biochemical discussion, several critical issues limit its impact. I have a few major concerns which needs to be addressed for better clarity.

Major Comments:

  1. The authors attempt to compare three models (canonical, ADP-privation, and GABA-shunt) but does not clearly define the fundamental biochemical discrepancies between them. For example, while the introduction outlines the importance of these models, their distinctions remain vague. Please clearly define the metabolic differences and the implications for β-cell function in the introduction.
  2. The authors strongly advocate for the ADP-privation model, but the evidence remains largely theoretical. For instance, while Lewandowski et al. demonstrated PKM2 activity in INS-1 cells, its physiological relevance in human β-cells remains unclear. Authors could present additional experimental support or acknowledge the limitations of the ADP-privation model more explicitly.
  3. The authors state that mitochondrial ATP production is dispensable for insulin secretion, yet discusses numerous findings showing mitochondrial metabolism’s role in β-cell function (e.g., tamoxifen-inducible β-cell-specific knockouts of respiratory complexes). Please address these contradictions and reconcile findings from different studies.
  4. There are conflicting statements regarding lactate production in β-cells. While some parts suggest β-cells rely on anaerobic glycolysis at low glucose concentrations, others emphasize mitochondrial oxidative phosphorylation as the dominant pathway. Provide a clear explanation of how β-cell metabolism shifts across glucose concentrations.
  5. The authors introduce the GABA-shunt model but does not explore its mechanistic details in depth. Expand the discussion on how the GABA-shunt model integrates with existing metabolic pathways.

Minor Comments:

  1. Please rephrase this sentence in the abstract for better clarity - "Analysis of precedent and new data should allow now and then to suggest modifications for a future integral view of the stimulus-secretion coupling of glucose-induced insulin secretion."
  2. Fig 1- Authors could provide a clearer legend and consider using a flowchart format to emphasize key differences among the three models.
  3. Please expand abbreviations when used for the first time such as ‘OxPhos’.
Comments on the Quality of English Language

There are some awkward sentences which could be rephrased, and the manuscript requires a check for the grammatical errors.

Author Response

(The authors gave the same response as above.)

Round 2

Reviewer 1 Report

Comments and Suggestions for Authors

The paper has been fixed according to the suggestions, and has been improved significantly. 

Reviewer 3 Report

Comments and Suggestions for Authors

The revised manuscript is an improvement over the previous version, but it still contains too many abbreviations. Full names should be clearly presented when first introduced to ensure clarity. (e.g. AQUI)
